# Multimodal Prehabilitation in Colorectal Cancer: Improving Fitness, Lifestyle, and Post-Surgery Outcomes

**DOI:** 10.3390/healthcare13091083

**Published:** 2025-05-07

**Authors:** María-Pilar Suárez-Alcázar, Ana Folch Ayora, María Muriach, Paula Recacha-Ponce, M.-Elena Garcia-Roca, Alba Coret-Franco, Juan Carlos Pastor-Mora, Pablo Salas-Medina, Eladio J. Collado-Boira

**Affiliations:** 1Nursing Department, University of Jaime I, Av. Vicente Sos Baynat s/n, 12071 Castellón de la Plana, Castellón, Spain; malcazar@uji.es (M.-P.S.-A.); afolch@uji.es (A.F.A.); psalas@uji.es (P.S.-M.); colladoe@uji.es (E.J.C.-B.); 2Medicine Department, University of Jaime I, Av. Vicente Sos Baynat s/n, 12071 Castellón de la Plana, Castellón, Spain; muriach@uji.es; 3Department of Physical Activity and Oncology, University of Jaime I, Av. Vicente Sos Baynat s/n, 12071 Castellón de la Plana, Castellón, Spain; garciroc@uji.es; 4Hospital Universitario General de Castellón, Av. Benicàssim, 128, 12004 Castellón de la Plana, Castellón, Spain; coret@uji.es (A.C.-F.); moraj@uji.es (J.C.P.-M.)

**Keywords:** prehabilitation, colorectal cancer, physical exercise, surgical complications, hospitalization

## Abstract

**Objectives**: This study aimed to analyze the effect of a multimodal prehabilitation program for colorectal cancer patients in body composition, physical and cardiorespiratory fitness as well as its ability to reduce postoperative complications. **Methods**: A longitudinal observational study evaluated the efficacy of a prehabilitation intervention based on four components: (a) health education and self-care, (b) nutritional counseling, (c) psychological support, and (d) supervised physical exercise. Body composition was determined through bioelectrical impedance analysis; physical fitness variables such as strength was measured by a handgrip dynamometer for upper limbs, and a squat–jump test, countermovement jump test using a contact platform, and a chair–stand test for lower limbs. Flexibility was assessed with the sit-and-reach test. Cardiorespiratory fitness was assessed with the 6 min walking test (6MWT). Moreover, we measured lifestyles related to the amount of physical exercise by accelerometry. **Results**: The final cohort included 30 patients. Patients completed an average of 9.90 ± 5.26 exercise sessions. Statistically significant changes with varying effect sizes were observed in the following outcomes: SJ values in cm and W for both sexes (*p* = 0.021/d = 0.14 and *p* = 0.043/d = 0.10, respectively), SJ in W for women (*p* = 0.023/d = 0.21), all chair-stand test values (*p* = 0.021/d = 0.65 for men, *p* = 0.004/d = 2.08 for women, and *p* = 0.000/d = 0.84 for both sexes), and sit-and-reach for both sexes (*p* = 0.005/d = 0.12) and for men (*p* = 0.044/d = 0.08). All 6MWT values had statistically significant changes (*p* = 0.001/0.46). Women reduced the weekly minutes spent in sedentary behavior (*p* = 0.037/d = 0.65) and increased the minutes spent performing light physical activity (*p* = 0.037/d = 0.63). With regard to surgical outcomes, there was a tendency towards a decrease in postoperative complications and hospitalization days, as well as minutes in postoperative REA (*p* = 0.009/d = 0.69) in relation to the control group. **Conclusions**: Participation in a multimodal prehabilitation program improves several aspects of physical condition and lifestyles related to the amount of physical exercise and reduces both days of hospitalization and several complications post-surgery.

## 1. Introduction

In oncology, prehabilitation refers to the period within the continuum of care that spans from diagnosis to the start of acute treatment. This phase incorporates physical and psychological interventions designed to evaluate and enhance the patient’s initial functional capacity [1,2,3,4,5,6,7].

Surgery is a cornerstone of curative treatment for patients with colorectal cancer; however, it carries a significant risk of morbidity [8,9] and postoperative mortality [8]. It is a stressful procedure that can lead to various types of post-surgery complications [8].

Prehabilitation encompasses all strategies and interventions [10] aimed at enhancing the patient’s functional capacity [8,11] before undergoing a stressful event such as surgery [8,12,13]. It improves the patient’s tolerance to the procedure [13,14], reduces the incidence and severity of related complications [10,14,15,16], and accelerates their recovery [11,12,14,15,17].

Numerous prehabilitation programs described in the literature include nutritional counseling [9,10,18,19,20,21,22,23,24], psychological counseling [10,12,13,14,20,21,22], cessation of toxic habits such as tobacco smoking as a “medical optimization” [8,9,12,13,18,19,22,24], and physical exercise [8,10,13,18,19,20,21,22,23,24]. However, current prehabilitation programs do not typically incorporate a four-modality intervention model. Our initiative is distinguished by being the first to be led by a nursing unit, representing a significant departure from the traditional roles attributed to nurses in prehabilitation. Previously, nursing involvement has primarily focused on specific components such as nutritional counseling [13], psychological support [25], or participant recruitment [26], rather than the overall coordination and management of the program.

The primary goal of nutritional counseling is to optimize nutrient reserves in the pre-operative period and provide adequate nutrition to counteract the catabolic response to surgery [27]. Outpatient nutritional screening is essential for all patients undergoing major surgery [27].

Psychologically, prehabilitation provides patients with the opportunity to gain greater control over their health outcomes while simultaneously reducing their anxiety levels. Evidence suggests that psychological screening and early interventions imple-mented immediately after diagnosis can improve the psychosocial adjustment of oncology patients [2].

In relation to health education and self-care, patient empowerment becomes a relevant factor, and it is essential to support them in utilizing their self-care abilities [28].

Nursing care for patients undergoing surgery focuses on their psychological needs, physical condition, and patient education regarding the procedures they will undergo [29] as well as the unhealthy behaviors they should avoid. In fact, existing guidelines for surgical patients strongly emphasize the importance of eliminating harmful habits like smoking [27].

The inclusion of physical exercise in prehabilitation programs extends beyond merely improving patients’ physical conditioning. Muscle plays a regulatory role in the body’s metabolic and inflammatory homeostasis [30].

Exercise in patients with colorectal cancer is considered a safe option that can provide significant benefits, including increased aerobic capacity, enhanced antioxidant capacity, improved insulin sensitivity, and a favorable shift in body composition toward greater lean mass relative to fat mass [24]. Training may also help reduce the number or severity of surgery-related complications [12] and improve functional recovery afterward [12], leading to a shorter hospital stay [27].

General conditioning exercises focus on strength [15], flexibility, aerobic training [15], and physical and cardiovascular fitness, both of which tend to decline in the postoperative period [2,24,27].

The use of triaxial accelerometry provides an objective measure to assess changes in lifestyle specifically related to the intensity and volume of daily physical activity.

However, few studies have referenced its use in this context [23,29,30,31,32], and those that do are highly heterogeneous in terms of study design.

It was hypothesized that participation in the Multimodal Prehabilitation Colorectal Cancer Program (a) enhances body composition, (b) improves physical and cardio-respiratory fitness, and (c) reduces the number and complexity of postoperative complications, days of hospitalization, and time in a postoperative resuscitation unit. The main objective of this research is to analyze the effect of a multimodal prehabilitation program for patients with colorectal cancer on body composition, physical and cardiorespiratory fitness, postoperative complications, days of hospitalization, and time in a resuscitation unit.

## 2. Materials and Methods

### 2.1. Design

This is a longitudinal study that assessed the effectiveness of a personalized prehabilitation program by comparing pre- and post-intervention outcomes. One month after surgery, postoperative variables were gathered and analyzed against those of a control group consisting of individuals with comparable characteristics who had undergone surgery the previous year. The control group’s postoperative data were obtained through a retrospective review of their medical records.

### 2.2. Sample

The study population is comprised of patients diagnosed with colon cancer who have a surgical indication as part of their therapeutic plan and are referred from the Surgical Services of Universitario General de Castellón Hospital (HUGCS).

#### 2.2.1. Selection Criteria

Inclusion criteria: Age over 18 years, diagnosis of colon cancer with a surgical indication as part of their therapeutic plan from HUGCS, and capacity to provide informed consent.

Exclusion criteria: Inability to understand provided information, insufficient knowledge of the Spanish language, inability to undergo the scheduled intervention due to physical condition, inability to be contacted, lack of transportation, surgery scheduled in less than one week, emergency surgery due to obstruction, perforation, hemorrhage, or similar reasons during the prehabilitation period, and neoadjuvant treatment before colon surgery. The control group comprised patients with colorectal cancer and primary surgery indication operated on in the previous year of the intervention, matched by sex and age (±5 years) with the intervention group patients.

#### 2.2.2. Sample Size Calculation

The sample size was estimated using the Fisterra tool [33]. To this end, the number of colorectal cancer patients who had undergone surgical intervention in the year prior to the study—91 patients—was considered. A 90% confidence level and a 15% replacement rate were also considered. Based on these parameters, the optimal sample size was estimated at 58 patients. To reach this sample size, all colorectal cancer patients scheduled for primary surgery from May 2023 to May 2024 were included in the study, provided they met the inclusion criteria and gave their informed consent to participate. As detailed in the results section, 80 patients were referred by the surgical department, but only 30 met the inclusion criteria and completed the intervention protocol, thereby comprising the final intervention cohort. Once this intervention group was established, an equal number of control patients were included in the study. These were identified retrospectively from the medical records of patients who had undergone surgery the previous year at the same hospital, and therefore had not participated in any prehabilitation program.

### 2.3. Intervention

The responsible surgeons facilitated contact between patients with the prehabilitation unit at University Jaume I. The nurse of the prehabilitation unit contacted the patients by phone who had requested information about the project and met them in person to give them information regarding the study, the form of data collection, and procedures to guarantee their anonymity and confidentiality. After that, written informed consent for voluntary participation was obtained. The prehabilitation program has been structured around four key interventions: (a) health education and self-care led by the nurses of the unit, (b) nutritional counseling provided by a nutritionist, (c) psychological support delivered by a psycho-oncologist, and (d) supervised physical exercise conducted by exercise specialists for cancer patients.

#### 2.3.1. Health Education and Self-Care

The health education and self-care intervention were carried out in a face-to-face session. During this session, participants received information about the postoperative instructions provided by the surgical team and information about the surgical procedure they underwent. They also received general information on the benefits of abstinence prior to surgery and the health risks associated with the consumption of substances such as tobacco smoking and alcohol, as well as nutritional information emphasizing a healthy diet with focus on adequate protein intake. Moreover, during this session, we conducted a screening of nutritional status and anxiety status of the patients.

#### 2.3.2. Nutritional Counseling

Patients identified through screening with nutritional risk were referred to specific nutritional counseling.

#### 2.3.3. Psychological Support

Patients identified through screening at risk of pathological anxiety were referred to a psycho-oncologist.

#### 2.3.4. Supervised Physical Exercise

The planned, supervised, and personalized exercise sessions took place twice a week, combining moderate-intensity aerobic exercise with strength training from the moment of study enrollment to the surgery date.

Each session began with a 10-min warm-up phase that included joint mobility and balance exercises to prepare the body for more intense activity. This was followed by a 40-min main workout segment designed to enhance both upper and lower body muscular strength as well as cardiorespiratory fitness. The session included a structured circuit of 8–12 functional exercises, such as squats, front and lateral lunges, abdominal crunches, calf raises, glute bridges, core stabilization exercises, biceps curls, shoulder presses, punches, jumping jacks, and stationary walking or jogging. The circuit was designed with two sets of 10–12 repetitions for strength-based exercises and 30-s intervals for aerobic activities. To ensure progressive overload, the training volume was gradually increased by modifying the number of repetitions, sets, and exercise complexity.

To conclude the session, the final 10 min were dedicated to cooling down, which involved stretching the primary muscle groups and incorporating breathing and relaxation techniques.

At the end of every session, perceived exertion was assessed using the CR-10 version of the Borg Scale [34]. The results were used to tailor the intensity of the upcoming sessions, aiming for a target exertion level between 6 and 8 points, corresponding to a moderate intensity level.

### 2.4. Outcomes

Outcomes included sociodemographic variables: age, sex, marital status, education level, and number of children. Body composition was determined through bioelectrical impedance analysis by Tanita BC-780MA (Tanita Corp., Tokyo, Japan) [35]. The variables obtained were weight, body mass index (BMI), % body fat, and muscle mass in kilograms. We measured the height with a portable stadiometer SECA 213 (Seca GmbH & Co. Kg, Hamburg, Germany) [36].

Anxiety levels were evaluated with the State-Trait Anxiety Inventory-State (STAI-S) [37], and nutritional screening was conducted using the Malnutrition Universal Screening Tool (MUST) [38,39].

Physical fitness was measured through the strength variables for upper-limb measured using a Camry handgrip dynamometer [40], lower-limb by squat-jump (SJ) test and counter movement-jump (CMJ) test using a contact platform Cronojump, BoscoSystem^®^ (Chronojump, Barcelona, Spain) [41,42], and the chair-stand test [43]. Handgrip strength is a widely used functional test for assessing strength and functional status in patients with cancer [44] and some similar tests to the chair-stand test were collected in the prehabilitation systematic review of Waterland et al. (2021) [45]. But, given the limited references found in the literature regarding the use of these tests in the context of prehabilitation, we decided to include the use of a contact platform to evaluate lower-limb strength.

Flexibility was registered with the sit-and-reach test [46]. Flexibility is generally not considered in prehabilitation research; nevertheless, flexibility is an important component of an individual’s functional capacity and represents another aspect that cancer patients will need to work on, especially if their treatment process includes, for example, chemotherapy [47]. Research indicates that certain drugs used in chemotherapy may cause adverse effects on the muscles, resulting in loss of muscle mass, strength, or flexibility [47].

Cardiorespiratory fitness was assessed with the 6 min walking test (6MWT) [48]. This test is commonly applied in the context of prehabilitation, provides a comprehensive assessment of the physiological demands in response to moderate physical activity [8], and has demonstrated validity in surgical populations [11,49]. We also measured lifestyles related to the amount of physical activity by accelerometry using GENEActiv (Activinsights Ltd. Kimbolton, Cambridgeshire, UK) portable wearables [50]. There is limited scientific literature regarding the use of accelerometry to evaluate daily physical activity during acute treatment in women with breast cancer, but digital devices like accelerometers are more accurate and quantifiable tools for measuring the level of physical activity [51,52,53] than questionnaires or category-based assessments of volume and intensity, which provide only broad estimates of daily physical activities [51,52,53,54].

One month after classification, hospitalization, and completion of surgery, postsurgical variables were collected and compared with a control group: type of surgery, patients with an ostomy and postoperative complications classified according to the Clavien-Dindo [55], days as well as minutes in a postoperative resuscitation unit.

### 2.5. Assessments

A baseline assessment was carried out after the inclusion of the patients in the program just after diagnosis. In this assessment, we measured sociodemographic variables, body composition, and physical and cardiorespiratory fitness variables, and patients were fitted with an accelerometer to measure their activity level during a week. Anxiety and malnutrition screening were also carried out.

Another preoperative assessment was carried out days before surgical intervention. We measured body composition, physical and cardiorespiratory fitness, and levels of anxiety. One week before this appointment, we fitted patients again with an accelerometer to measure their activity level for a week.

Finally, one month after completion of surgery, surgical variables (postoperative complications, hospitalization days, and time in a postoperative resuscitation unit) were collected from their medical history and compared with a control group.

### 2.6. Statistical Analysis

Statistical analyses were conducted using IBM SPSS Statistics for Windows, version 29.0 (IBM Corp., Armonk, NY, USA), considering a *p*-value < 0.05 as the threshold for statistical significance. Given the sample size, the Shapiro–Wilk test was applied to assess the normality of the variables. Since the data did not meet parametric assumptions, non-parametric tests were utilized.

Descriptive statistics were reported as mean and standard deviation for continuous variables, while categorical variables were summarized using frequencies and percentages. To evaluate changes before and after the intervention, Wilcoxon tests were performed. Additionally, effect size was calculated using Cohen’s d and classified as follows: very small (d ≤ 0.1), small (d ≤ 0.2), medium (d ≤ 0.5), large (d ≤ 0.8), very large (d ≤ 1.2), and huge (d ≥ 2) [56,57].

The chi-square test was applied to determine significant differences in surgical complications, while the Mann–Whitney U test was used to compare hospitalization and time in a postoperative resuscitation unit.

### 2.7. Ethical Considerations

All participants signed informed consent forms. Ethical approval was obtained from the Universitat Jaume I (CEISH/87/2023; 30 November 2023) and HUGCS Research and Ethics Committee (PREHAB_2023; 24 April 2023), and the project was registered on ClinicalTrials.gov (NCT05887531).

## 3. Results

### 3.1. Participants and Sociodemographic Characteristics

From the hospital, 80 patients were referred to the prehabilitation program. A total of 44 of these patients declined to participate in the study or did not meet the inclusion criteria in the study: seven had physical limitations, two had surgical interventions in one week, one had an emergency ostomy, six did not have possibility of moving and 27 withdrew their participation. Of those accepted (n = 36), we suffered six losses during the process. Finally, 30 patients completed all our assessments and were included in the prehabilitation program. Figure 1 presents the flowchart outlining the patient selection process.

Table 1 shows sociodemographic characteristics of the sample. The study sample was homogeneous, showing no statistically significant differences in these variables.

### 3.2. Anthropometry and Body Composition

A significant reduction in % fat is observed in men, though the effect size is negligible (Table 2).

### 3.3. Anxiety and Malnutrition Screening

No patient scored above 40 points for classifying clinically significant symptoms of anxiety by the STAI-S, and only one of the patients needed to be referred to a nutritionist with a medium risk of malnutrition determined by the screening.

### 3.4. Physical and Cardiorespiratory Fitness

Our patients completed an average of 9.90 ± 5.26 exercise sessions. As shown in Table 3, statistically significant changes were observed in several variables of physical and cardiorespiratory fitness: SJ in cm and W for both sexes (*p* = 0.021/d = 0.14 and *p* = 0.043/d = 0.10, respectively), SJ in W for women (*p* = 0.023/d = 0.21), all chair-stand values (*p* = 0.021/d = 0.65 in men, *p* = 0.004/d = 2.08 in women, and *p* = 0.000/d = 0.84 for both sexes), sit and reach for both sexes (*p* = 0.005/d = 0.12) and specifically for men (*p* = 0.044/d = 0.08), as well as all 6MWT values (*p* = 0.044/d = 0.34 in men, *p* = 0.008/d = 0.60 in women, and *p* = 0.001/d = 0.46 for both sexes).

### 3.5. Minutes According to the Activity Level of the Participants

Table 4 shows the weekly minutes according to the activity level of the participants. Women reduced the weekly minutes spent in a sedentary state (*p* = 0.037/d = 0.65) and increased the minutes engaged in light physical activity (*p* = 0.037/d = 0.63). The remaining data were not statistically significant.

### 3.6. Surgical Variables

As far as surgical complications are concerned, twice as many patients had surgical complications in the control group (n = 6) than in the intervention group (n = 3), but this difference was not statistically significant.

Three patients in the intervention group experienced Grade II surgical complications. One patient developed a fever due to colon inflammation, and another experienced diarrhea; both required antibiotic treatments. Another patient received a continuous phenylephrine infusion due to hypotension in the immediate postoperative period.

However, in the control group, six patients experienced complications. Two of them had Grade I complications, two patients had Grade II complications, and two more experienced complications classified as Grade III, specifically Grade IIIb, as they required reoperation under general anesthesia (one case involved an anastomotic dehiscence following the initial surgery, while the other presented with abdominal evisceration).

Finally, Table 5 presents data on the length of hospital stay, time spent in the resuscitation unit, type of surgery performed, and the presence or absence of an ostomy.

## 4. Discussion

Altogether, the results of this study show that colorectal cancer patients enrolled in our prehabilitation program exhibited an improvement in cardiorespiratory fitness from the baseline to the preoperative assessment. Moreover, lower limb strength increased, as evidenced by enhanced performance in the chair-stand test, a finding further corroborated by the countermovement jump test results. Finally, all participants showed enhanced flexibility.

It is noteworthy that the primary aim of this study is to evaluate, on one hand, the impact of a multimodal prehabilitation program on body composition in patients with colorectal cancer. However, despite observing statistically significant results in body fat percentage in men, we consider the effect size to be minimal and insufficient to claim clinical relevance.

On the other hand, we aimed to assess whether the program had any effect on physical and cardiorespiratory fitness. In relation to physical fitness, the findings of our study regarding grip strength were favorable. However, they did not reach statistical significance. This agrees with some of the studies in the literature. However, in all cases reviewed, the improvement was always compared to a control group, whereas in our study, the effects of prehabilitation were assessed using a pre–post intervention design. For example, in a study by Bojesen et al. (2023) [19], the prehabilitation group achieved grip strength scores of 29.7 kg compared to 23.1 kg in the control group; however, their results were not statistically significant, and they did not report separate data for each upper limb. Similarly, in a study conducted by Northgraves et al. (2019) [58], grip strength improved by 1.4 kg in the right hand and 0.2 kg in the left hand, but the results were also not statistically significant. Unlike our study, this intervention was unimodal, focusing exclusively on physical exercise. Regarding strength in lower limbs measured by the SJ test and the CMJ test using a contact platform, values for both sexes showed statistical differences in cm and W and SJ in W for women too. However, we did not find bibliographic references on its use in oncology patients for comparison with our results.

In this same line, chair-stand values were positives, and statistical differences were shown in all the results (both sexes, male and female). Data in the literature refer to similar tests such as the time up and go (TUG), stair climbing test (SCT), or five time sit to stand (FTSTS) [59]. Northgraves et al. (2019) [58], for example, implemented a unimodal intervention consisting only of physical exercise. Although their study was quite like ours in terms of sample size, mean age, type of exercise, supervision, and duration, they reported a non-significant improvement in the TUG and SCT tests, while no improvement was observed in the FTSTS test. Dronkers et al. (2010) [60], which implemented a trimodal intervention with characteristics like our study, reported an improvement in the SCT test, although it was not statistically significant.

Flexibility measured by the sit-and-reach test increased for both sexes, and especially for men. We have not found any references in the literature discussing flexibility within the context of prehabilitation. However, we decided to include this parameter in the prehabilitation program due to the fact that a significant number of these patients will undergo chemotherapy in the future. As we mentioned in the methodology, studies suggest that specific chemotherapy drugs can have negative impacts on flexibility [47].

The results regarding cardiorespiratory fitness measured by 6MWT have been positive. The patients have improved the values measured between the baseline assessment and the preoperative assessment, and the data were also statistically significant. Authors such as Bogani et al. (2020) [61] suggest that aerobic exercise may be beneficial for improving cardiopulmonary function around the time of surgery and for enhancing tissue oxygenation.

Our findings are consistent with those reported by Li et al. (2013) [12], who conducted a pilot study involving a sample of 42 participants with a mean age of 67.4 years. Their intervention employed a trimodal prehabilitation approach, combining moderate-intensity exercise with strength training, resulting in an improvement of 42 m between the preoperative and baseline assessments (*p* < 0.01). Also, in a study by Fulop et al. (2021) [62] where, as in our case, they combined moderate-intensity aerobic exercise with strength training, the results were statistically significant (*p* < 0.001). Their sample consisted of 89 patients with a mean age of 70 years, and the intervention lasted between 3 and 6 weeks. It is important to note that both the mean age and the duration of the intervention were higher than in our study. In the randomized clinical trial conducted by Waller et al. (2022) [23], a sample of 11 patients with a mean age of 55.5 years underwent a trimodal prehabilitation intervention combining aerobic exercise with strength training. The improvement in meters walked between the preoperative and baseline assessments was substantial, specifically 85 m. However, the average time from inclusion in the program to surgery was 30.5 days, with exercise sessions performed 5 days per week, which was three times higher than our average exercise frequency per patient (9.90 ± 5.26).

Referring to the monitoring of patients through accelerometry, it was revealed that in women, there was significant decrease in daily sedentary time and a significant increase in light physical activity levels. In the literature related to prehabilitation, there are few references to the use of accelerometers as an objective measure to assess the intensity and volume of physical activity in patients. The study most similar to ours in terms of the prehabilitation model, which combines moderate-intensity physical exercise with strength training, is the one conducted by Waller et al. (2022) [23]. However, they compared a control group with an intervention group and found that the minutes spent performing vigorous physical activity were significantly higher in the intervention group compared to the control group, with these differences being statistically significant (*p* = 0.02). A similar trend was observed for moderate-intensity activity, although the results did not reach statistical significance, while light-intensity activity levels were comparable between both groups. It is important to note that this study utilized a Fitbit Smartwatch, which allowed patients to monitor their physical activity levels in real-time, a factor that we believe may have motivated participants to increase their activity levels. Additionally, the average time to surgery for participants in this study was 30.5 days, with physical exercise performed 5 days per week, representing a significantly longer program compared to ours, which had an average of 9.90 days of physical exercise per patient.

Finally, we sought to investigate the effect of the prehabilitation program on postoperative complications, days of hospitalization, and time in resuscitation unit. The number and complexity of complications post-surgery and days of hospitalization decreased in the prehabilitation group compared to the control group. However, this reduction was not statistically significant. Nevertheless, a statistically significant reduction in the time spent in the resuscitation unit was observed among patients who underwent the prehabilitation intervention compared to those in the control group, despite both groups being managed according to the same standardized resuscitation protocol. This finding has not been documented in any of the studies identified in the existing literature, thus precluding direct comparisons with previous research. Although the control group included a higher proportion of non-laparoscopic procedures, it should be noted that this group also underwent fewer ostomies. Therefore, we do not consider these differences sufficient to explain the shorter resuscitation unit stay observed in the intervention group.

Regarding hospitalization days, several studies in the literature indicate no significant differences in the length of hospital stay between intervention and control groups [52]. The case of Carli et al.’s (2020) [25] study with a sample of 55 patients with a mean age of 78 years and an intervention of similar approach lasting four weeks did not identify differences between groups. The study of Gillis et al. (2014) [63] with a sample of 38 patients with a mean age of 65.7 years and with a physical exercise intervention like ours also found no statistically significant differences between groups.

Concerning post-surgical complications, several references in the literature report results favoring prehabilitation interventions, although these were not statistically significant or clinically relevant. For example, in a study by López-Rodríguez-Arias et al. (2022) [64], complications in the prehabilitation group were lower than in the control group, although it was not significant. In the study conducted by Bojesen et al. (2023) [19], an intervention with a trimodal prehabilitation approach was conducted in a sample of 16 patients with a mean age of 80 years, and supervised high-intensity exercise sessions were carried out for four weeks. Along the same lines, the findings of Carli et al. (2020) [25] align with our results. Their study implemented a similar intervention to ours; however, although the intervention group experienced fewer complications than the control group, the difference was not statistically significant.

### Strengths and Limitations of the Study

The main limitation of our study was the sample size, which restricts the generalizability of the findings. Given the limited number of participants, caution is warranted when extrapolating the results to broader populations. To enhance the feasibility of similar studies in the future, it is essential to explore and optimize recruitment strategies, ideally incorporating qualitative assessments of patient perspectives. Such approaches could facilitate greater participation by identifying and addressing barriers to enrollment in prehabilitation programs.

On the other hand, current prehabilitation interventions exhibit considerable heterogeneity in terms of duration, intensity, setting (home-based vs. hospital-based), and modality (unimodal vs. multimodal), as well as in the specific components related to exercise, nutrition, psychological support, outcome indicators, and the measurement instruments employed. This variability poses significant challenges when attempting to compare outcomes across studies. Therefore, future research should aim to identify the most effective elements of prehabilitation, including the optimal type and intensity of exercise, the appropriate setting, and the ideal duration. Moreover, the standardization of outcome measures is crucial to facilitate more accurate and meaningful meta-analyses.

## 5. Conclusions

Participants in our prehabilitation program did not show improvement in body composition, but exhibited enhanced physical and cardiorespiratory fitness. Subjects significantly improved their lower-limb strength, and all individuals showed gains in flexibility levels. Furthermore, a significant improvement was also demonstrated in the 6MWT. Furthermore, female colorectal cancer patients who participated in our prehabilitation program increased the volume and intensity of their routine physical activities, reducing the time spent in sedentary behavior and increasing the minutes of light-intensity activity. This could indicate a slight change in their lifestyle.

We also confirm that, following prehabilitation intervention in colorectal cancer patients, although without significance, there is a trend towards a reduction in the number and complexity of surgical complications and length of hospital stays. Additionally, the time patients spent in the resuscitation unit was significantly reduced.

## Figures and Tables

**Figure 1 healthcare-13-01083-f001:**
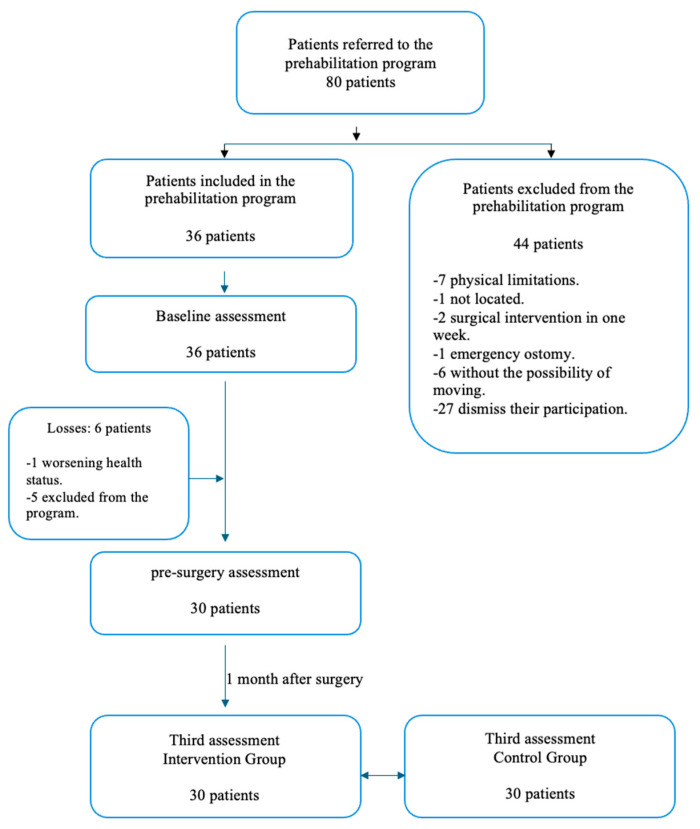
Description of patient selection.

**Table 1 healthcare-13-01083-t001:** Sociodemographic and clinical variables of intervention group.

	Male		Female		Both Male and Female		*p*-Value
	n	%	n	%	n	%	
Sex	17	56.7%	13	43.3%	30	100%	
Marital status							0.125
Single	3	17.6%	1	7.7%	4	13.3%	
Married or in a relationship	13	76.5%	8	61.5%	21	70.0%	
Widowed			2	15.4%	2	6.7%	
Separated or divorced	1	5.9%	2	15.4%	3	10%	
Study level							0.397
Primary	7	41.2%	7	53.8%	14	46.7%	
High-School	7	41.2%	5	38.5%	12	40.0%	
University	3	17.6%	1	7.7%	4	13.35%	
Employment Status							
Employed	5	29.4%	0	0%	5	16.7%	0.330
Unemployed	12	70.6%	13	100%	25	83.3%	
	**mean ± sd**		**mean ± sd**		**mean ± sd**		***p*-Value**
Age (years)	66.53 ± 8.01		62.54 ± 10.1		64.8 ± 9.08		0.240
Sons (number)	1.47 ± 0.94		1.31 ± 0.75		1.4 ± 0.85		0.614
Height (meters)	1.70 ± 0.55		1.55 ± 0.06		1.63 ± 0.09		

Values are presented in numbers and percentages for categorical variables and mean and deviation (sd) for continuous variables.

**Table 2 healthcare-13-01083-t002:** Anthropometric assessment.

VARIABLES	Male			Female			Both Male and Female		
	Pre	Post	*p*-Value/d	Pre	Post	*p*-Value/d	Pre	Post	*p*-Value/d
	Mean ± sd	Mean ± sd		Mean ± sd	Mean ± sd		Mean ± sd	Mean ± sd	
Weigh (kg)	83.32 ± 11.51	83.19 ± 11.89	0.917	65.37 ± 15.80	65.57 ± 15.91	0.327	75.55 ± 16.07	75.56 ± 16.17	0.647
BMI (kg/m^2^)	28.24 ± 4.10	28.52 ± 4.17	0.394	27.37 ± 7.20	27.44 ± 7.20	0.412	27.86 ± 5.56	28.05 ± 5.60	0.526
% body fat	29.62 ± 8.88	28.69 ± 8.89	**0.028/0.10**	34.93 ± 9.64	34.90 ± 9.32	0.944	31.92 ± 9.44	31.38 ± 9.45	0.054
Muscle mass (kg)	32.47 ± 4.85	32.57 ± 4.95	0.736	21.79 ± 3.90	21.86 ± 3.87	0.581	27.84 ± 6.94	27.93 ± 6.99	0.616

Values are presented as mean ± deviation (sd). Pre: variables collected before the intervention and Post: variables collected after the designed prehabilitation intervention. *p*-value < 0.05 indicates the existence of statistically significant differences between pre- and post-intervention values. d denotes the effect size. Bold indicates statistically significant values.

**Table 3 healthcare-13-01083-t003:** Physical and cardiorespiratory fitness.

	Male			Female			Both Male and Female		
	Pre	Post	*p*-Value/d	Pre	Post	*p*-Value/d	Pre	Post	*p*-Value/d
	Mean ± sd	Mean ± sd		Mean ± sd	Mean ± sd		Mean ± sd	Mean ± sd	
Strength (kg)									
Right	37.71 ± 9.19	38.87 ± 8.71	0.109	22.48 ± 3.39	22.62 ± 3.26	0.624	31.11 ± 10.50	31.83 ± 10.65	0.120
Left	35.12 ± 9.09	35.68 ± 7.94	0.378	20.50 ± 4.32	20.63 ± 4.49	0.972	28.82 ± 10.35	29.16 ± 10.04	0.502
SJ (cm)	14.17 ± 6.13	14.85 ± 5.33	0.121	8.66 ± 1.97	9.58 ± 2.52	0.099	11.89 ± 5.56	12.67 ± 5.07	**0.021/0.14**
SJ (W)	699.13 ± 182.72	679.21 ± 148.74	0.469	402.18 ± 182.72	435.05 ± 108.81	**0.023/0.21**	558.67 ± 203.10	578.18 ± 169.74	**0.043/0.10**
CMJ (cm)	15.66 ± 6.69	15.96 ± 5.91	0.569	10.14 ± 2.18	9.89 ± 2.73	0.638	13.37 ± 5.92	13.45 ± 5.67	0.802
CMJ (W)	701.47 ± 190.36	1043.21 ± 1363.64	0.234	513.51 ± 266.24	439.54 ± 98.455	0.209	623.69 ± 239.64	793.43 ± 1076.08	0.855
Chair-Stand	19.47 ± 3.04	21.71 ± 3.73	**0.021/0.65**	28.31 ± 3.47	31.71 ± 2.83	**0.004/2.08**	18.97 ± 3.23	21.73 ± 3.32	**0.000/0.84**
Sit & Reach (cm)	−3.22 ± 11.62	−2.20 ± 11.27	**0.044/0.08**	1.34 ± 9.91	3.06 ± 8.34	0.080	−1.13 ± 10.95	0.23 ± 10.19	**0.005/0.12**
6MWT (m)	544.71 ± 89.03	571.82 ± 68.23	**0.044/0.34**	505.23 ± 81.91	563.31 ± 109.47	**0.008/0.60**	527.6 ± 86.60	568.13 ± 86.86	**0.001/0.46**

Values are presented as mean ± deviation (sd). Pre: variables collected before the intervention and Post: variables collected after the designed prehabilitation intervention. *p*-value < 0.05 indicates the existence of statistically significant differences between pre- and post-intervention values. d denotes the effect size. Bold indicates statistically significant values.

**Table 4 healthcare-13-01083-t004:** Weekly minutes according to the activity level of the participants.

	Pre (Mean ± sd)	Post (Mean ± sd)	*p*-Value/d
Sedentary			
Both sexes	5981.56 ± 1289.70	5958.61 ± 1359.30	0.744
Male	5382.25 ± 1365.20	6396.50 ± 882.70	0.093
Female	6461.00 ± 992.95	5608.30 ± 1557.69	**0.037/0.65**
Light			
Both sexes	3289.61 ± 1160.15	3288.28 ± 1331.05	0.744
Male	3810.00 ± 1288.33	2794.00 ± 644.32	0.093
Female	2873.30 ± 839.59	3683.70 ± 1582.76	**0.037/0.63**
Moderate			
Both sexes	784.67 ± 356.60	811.94 ± 354.78	0.248
Male	841.75 ± 283.63	848.13 ± 299.29	0.889
Female	739.00 ± 399.80	783.00 ± 391.18	0.169
Vigorous			
Both sexes	24.173 ± 71.15	21.17 ± 64.44	0.073
Male	46.00 ± 102.27	41.38 ± 92.49	0.173
Female	6.70 ± 79.59	5.00 ± 6.69	0.206

Values are presented as mean ± deviation (sd). Pre: variables collected before the intervention and Post: variables collected after the designed prehabilitation intervention. *p*-value < 0.05 indicates the existence of statistically significant differences between pre- and post-intervention values. d denotes the effect size. Bold indicates statistically significant values.

**Table 5 healthcare-13-01083-t005:** Surgical variables.

	Control (Mean ± sd)		Intervention (Mean ± sd)		*p*-Value/d	
Hospitalization (days)	6.14 ± 4.98		5.45 ± 1.80		0.607	
Time in Resuscitation Unit (min)	1488.08 ± 563		1118.77 ± 504		**0.009/0.69**	
	**Control (n (%))**			**Intervención (n (%))**		
Type of Surgery						
Laparoscopy	25 (83.3%)			30 (100%)		
Laparotomy	2 (6.6%)			-		
Conversion from laparoscopy to laparotomy	3 (10.1%)			-		
Ostomy						
Yes	1 (3.4%)			7 (30%)		
No	29 (96.6%)			23 (70%)		

Values are presented as mean ± deviation (sd). *p*-value < 0.05 indicates the existence of statistically significant differences between pre- and post-intervention values. d denotes the effect size. Bold indicates statistically significant values.

## Data Availability

For ethical reasons related to the preservation of patient identity, the data presented in this study are available upon request to the corresponding author.

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
