# Peer review of "Multimodal Prehabilitation in Colorectal Cancer: Improving Fitness, Lifestyle, and Post-Surgery Outcomes"

_healthcare, 2025, doi:10.3390/healthcare13091083_

Round 1

Reviewer 1 Report

Comments and Suggestions for Authors

This manuscript describes a longitudinal study examining the efficacy of an intervention designed to optimize patients before surgery for colorectal cancer. 

Several issues are critical and need attention:

Language revision is imperative as it is difficult to interpret the precise meaning throughout the manuscript. Also the use of abbreviations needs writing in full the first time. 

Introduction: The knowledge gap needs to be described and the rationale for all elements of the intervention argued. The flexibility component has not been described, for example. 

Materials and methods: 

  • the recruitment methods, the intervention and the matching procedure for the comparison group needs to be described in depth. For example, how were patients approached, by whom? How was the intervention delivered? Who provided the interventions (which health care professionals?), what was the duration and contents of the interventions. I.e. how was progression in the exercise intervention operationalised? Here it is recommended to use the FITT principles  to describe the Frequency, Intensity, Type and Time of the exercises. The duration of the intervention is also not desribed.
  • The sample size calculation is poorly described and not logical. 
  • The choice of physical tests used has not been sufficiently argued, and seeing as the authors recognize that some of these tests have not previously been used in exercise studies for this patient group, it needs further attention.

Results: Table one is somewhat scant, and more patient characteristics for both groups would have been good. 

Discussion: Lacking an overview of the results to begin with. Further, the discussion lacks depth and structure, but it may also be clouded by the language issue. It needs to also adress the strengths and limitations of this study and indicate directions for future research.

Comments on the Quality of English Language

Language here is critical. I already commented on it above.

Author Response

Comments 1: Language revision is imperative as it is difficult to interpret the precise meaning throughout the manuscript. Also the use of abbreviations needs writing in full the first time. 

Response 1: We have sent the manuscript to an external translator to improve the linguistic level.

Comments 2: Introduction: The knowledge gap needs to be described and the rationale for all elements of the intervention argued. The flexibility component has not been described, for example. 

Response 2: We have written a paragraph in the introduction to describe the knowledge gap  (lines 62-68) and have justified the rationale for the elements in the methodology (lines 207-213)(lines 220-223)(line 225-230). The flexibility component has been better described (lines 214-219).

Comments 3: The recruitment methods, the intervention and the matching procedure for the comparison group needs to be described in depth. For example, how were patients approached, by whom? How was the intervention delivered? Who provided the interventions (which health care professionals?), what was the duration and contents of the interventions. I.e. how was progression in the exercise intervention operationalised? Here it is recommended to use the FITT principles  to describe the Frequency, Intensity, Type and Time of the exercises. The duration of the intervention is also not described.

Response 3: The description of the recruitment methods, the intervention and the matching procedure for the comparison group has been made improved (lines 138-144)(lines 148-156). We included a figure of the flowchart outlining of patient selection process (line 278). And following the reviewer's recommendations, we have used the FITT principles to describe the frequency, intensity, type, and time of the exercise (lines 177-180)(lines 187-193).

Comments 4: The sample size calculation is poorly described and not logical. 

Responses 4: We have attempted to describe again how the sample size calculation was performed and have corrected a detected error (line 132-144).

Comments 5:The choice of physical tests used has not been sufficiently argued, and seeing as the authors recognize that some of these tests have not previously been used in exercise studies for this patient group, it needs further attention.

Response 5:We have justified the choice of the tests used (lines 207-213)(lines 214-219)(lines 220-223)(line 225-230).

Comments 6: Results: Table one is somewhat scant, and more patient characteristics for both groups would have been good.

Response 6: We appreciate the reviewer's suggestions and will take them into account for future research. However, we have included, though not in the table of sociodemographic variables but in the surgical variables, whether the surgery was open or not and whether the patients ultimately had ostomies.

Comments 7: Discussion: Lacking an overview of the results to begin with (lines 329-334). Further, the discussion lacks depth and structure, but it may also be clouded by the language issue. It needs to also adress the strengths and limitations of this study and indicate directions for future research (lines 443-459).

Response 7: We have provided an initial summary of the relevant results (lines 331-336) and improved the structure of the discussion for better understanding (lines 337-343)(lines 370-373)(lines 414-415). Additionally, we have included a section on limitations and future research directions (lines 446-461). We have also modified the structure of the conclusion to make it clearer (lines 464-475).

Reviewer 2 Report

Comments and Suggestions for Authors

I have read with interest the paper about multimodal prehabilitation and colorectal surgery.

In general, English languaje should be revised. (Ie. Table 1: high (should be height))

Line 90  "Authors such as Bongani et al. (2020) [28] suggest that aerobic exercise may be beneficial for improving cardiopulmonary function at the time of surgery and enhancing tissue oxygenation." I would recomend to be place in discussion 

Table 1: marital status seems not to be essential in this study. I would suggest “living alone / vs living with people at Home)

Surgical variables should be revisited: no data about laparotomy vs laparoscopy, no data on bowel diversion (ilesotomy diversion).  I would also suggest to clarify if the Protocol center regarding  resuscitation unit is the same or it has change, as seems to much diferences in minutes. Could it be a change on clinical criteria? Also, clavien dindo grade 3 complications should be named, in both groups, as they clearly impact in the results of the study. 

In conclusions, i would suggest to change “...there is a clear trend toward a reduction in number and complexity of surgical  complications and length of hospital stay.” and write "... although without significance, there is a trend towards reduction in ..."

Comments on the Quality of English Language

English should be revisited. 

Author Response

Comments 1: In general, English languaje should be revised. (Ie. Table 1: high (should be height))

Response 1: We have sent the manuscript to an external translator to improve the linguistic level.

Comments 2: Line 90  "Authors such as Bongani et al. (2020) [28] suggest that aerobic exercise may be beneficial for improving cardiopulmonary function at the time of surgery and enhancing tissue oxygenation." I would recommend to be place in discussion 

Response 2: We have moved this reference to the discussion, as suggested by the reviewer.

Comments 3: Table 1: marital status seems not to be essential in this study. I would suggest “living alone / vs living with people at Home)

Response 3: We thank the reviewer for their comment, but the data were collected in this way, and subsequently, we do not know, for example, whether a single person lives alone or not, so it is not possible for us to make the suggested change.

Comments 4: Surgical variables should be revisited: no data about laparotomy vs laparoscopy, no data on bowel diversion (ilesotomy diversion).  I would also suggest to clarify if the Protocol center regarding  resuscitation unit is the same or it has change, as seems to much diferences in minutes. Could it be a change on clinical criteria? Also, clavien dindo grade 3 complications should be named, in both groups, as they clearly impact in the results of the study.

Response 4: We have reviewed the surgical variables and included whether the surgery was performed by laparotomy or laparoscopy, as well as whether the patients ultimately required bowel diversion (Table 5 Cont and line 328). We have tried to make the protocol in the resuscitation unit clearer (lines 418-426) and have also included which grade three complications the patients experienced (325-326).

Comments 5: In conclusions, i would suggest to change “...there is a clear trend toward a reduction in number and complexity of surgical  complications and length of hospital stay.” and write "... although without significance, there is a trend towards reduction in ..."

Response 5: We have implemented the reviewer's suggestion